# Cannabinoids and Extracellular Vesicles as Potential Biomarkers and Therapeutic Targets in Neuropsychiatric Disorders: A Hypothesis-Driven Review

**DOI:** 10.3390/ph18121817

**Published:** 2025-11-28

**Authors:** Bruno L. Marques, Pedro H. C. Lirio, Maria A. Vicente, Paula Unzueta-Larrinaga, Leyre Urigüen, Alline C. Campos

**Affiliations:** 1Department of Pharmacology, Ribeirão Preto Medical School, University of São Paulo, 3900, Bandeirantes Avenue, Monte Alegre, Ribeirão Preto 14049-900, SP, Brazil; bruno.lemesm@gmail.com (B.L.M.); pedrohcassaro@usp.br (P.H.C.L.); madrivic@gmail.com (M.A.V.); 2Center for Cannabinoid Research, Ribeirão Preto Medical School, University of São Paulo, 2650-Tenente Catão Roxo, Monte Alegre, Ribeirão Preto 14040-900, SP, Brazil; 3Department of Pharmacology, Universidad del Pais Vasco-Euskal Herriko Unibertsitate UPV/EHU, 48940 Leioa, Spain; paula_unzueta001@ehu.eus (P.U.-L.); leyre.uriguen@ehu.eus (L.U.); 4Biobizkaia Health Research Institute, 48903 Barakaldo, Spain; 5Centro de Investigacin Biomédica en Red de Salud Mental CIBERSAM, 28029 Madrid, Spain

**Keywords:** extracellular vesicles, neuropsychiatric disorders, cannabinoids, biomarkers

## Abstract

**Background and Objectives**: Neuropsychiatric disorders pose a major global health challenge, marked by high prevalence, limited diagnostic precision, and suboptimal therapeutic outcomes. Current diagnoses remain primarily clinical, lacking objective biomarkers, while many patients experience poor remission rates and frequent relapse. The endocannabinoid system (ECS), a central regulator of synaptic plasticity, neuroinflammation, and stress responses, is increasingly implicated in depression, anxiety, schizophrenia, and neurodegenerative diseases. In parallel, extracellular vesicles (EVs) have emerged as critical mediators of intercellular communication and promising biomarker sources, as they reflect the physiological or pathological status of their cells of origin. This review examines the hypothesis that interactions between ECS signaling and EV-mediated communication form a convergent pathway shaping vulnerability and resilience in neuropsychiatric disorders, with potential implications for biomarker identification and therapeutic innovation. **Methods**: This hypothesis-driven review was developed using a narrative approach, focusing on the interface between cannabinoids and EVs in neuropsychiatric conditions. Relevant publications were identified through PubMed, Scopus, and Web of Science searches up to September 2025. **Results**: Emerging evidence indicates a bidirectional relationship between ECS activity and EV biology: endocannabinoids can be loaded into EVs to facilitate intercellular signaling, while phytocannabinoids such as THC and CBD can alter EV release and cargo composition. **Conclusions**: We propose a hypothesis-driven framework in which the possible interplay between cannabinoids and EVs may stimulate new research and support the development of biomarker-guided, personalized therapeutic strategies for neuropsychiatric disorders.

## 1. Current Challenges for the Treatment of Neuropsychiatric Disorders

According to the World Health Organization (WHO), 1 in every 8 people (~1 billion) in the world is currently living with neuropsychiatric disorders, including both mental and neurological conditions [1]. They represent a significant challenge in global public health, with recent data highlighting high prevalence rates. In parallel, neurological conditions affect more than 3 billion people and have become the leading cause of disability and disease burden worldwide, with an 18% increase in global burden since 1990 [2].

Beyond their high prevalence, the impact of neuropsychiatric disorders is profound: these conditions rank among the leading causes of years lived with disability and loss of quality of life, surpassing cardiovascular diseases, cancer, and diabetes in terms of non-fatal disease burden [1,2].

Nevertheless, access to diagnosis and treatment remains limited. It is estimated that more than 75% of individuals with neuropsychiatric disorders do not receive adequate care, particularly in low- and middle-income countries, where there are significant disparities in access to specialized professionals and services [3,4,5].

Diagnosis of these conditions remains predominantly clinical, based on behavioral criteria and psychometric reports, contributing to diagnostic variability and hindering early and accurate identification. The lack of objective biomarkers, together with the genetic and molecular complexity of neuropsychiatric disorders, underscores the urgent need for new approaches to diagnosis and risk stratification [6,7].

Despite advances in pharmacological and psychosocial therapies, the global burden of neuropsychiatric disorders has not significantly decreased in recent decades, highlighting the limitations of current treatments and the pressing need for innovation in prevention, diagnostic, and intervention strategies [2,8].

Indeed, current treatments are characterized by limited remission rates and a significant risk of relapses. For instance, in major depressive disorders, response rates to first-line antidepressant treatment typically range from 50% to 65%, with remission rates being even lower. The STAR*D (Sequenced Treatment Alternatives to Relieve Depression) study, one of the largest and most comprehensive clinical trials in depression, demonstrated that remission rates decrease substantially with each successive treatment attempt: while the initial remission rate was 37%, it dropped to just 13% after four successive treatment steps [9,10].

A similar challenge is seen in neurodevelopmental disorders such as schizophrenia. Long-term studies report that up to 80% of individuals relapse within five years of their first psychotic episode, even with adherence to antipsychotic medication [11,12]. Moreover, hospital readmission rates remain high, with only 10–25% of patients remaining free from subsequent admissions over extended follow-up periods [13,14].

When considering anxiety disorders, the limitations of current treatments become equally evident. Clinical trials report response rates to pharmacological treatment ranging from 40% to 70%, while remission rates are typically lower, varying between 20% and 47% [15]. For instance, a large naturalistic study of patients with comorbid depression and anxiety treated with escitalopram found remission rates of 7.9% at week two, 38.8% at week eight, and 66.3% at week sixteen [16]. These findings are consistent with other studies showing that even after several months of optimized therapy, a significant proportion of patients do not achieve full remission and remain symptomatic.

In summary, while current therapies can lead to significant symptom reduction in neuropsychiatric disorders, complete and sustained remission remains elusive for a substantial subset of patients. This underscores the urgent need for objective biomarkers and innovative therapeutic strategies to improve clinical outcomes in patients.

In this context, in the present review, we introduce our hypothesis that the interface between cannabinoids and extracellular vesicles (EVs) represents a critical and underexplored axis in the pathophysiology of neuropsychiatric disorders. Specifically, we propose that cannabinoid–EV interactions may serve as a source of novel biomarkers and therapeutic targets, offering a promising avenue for improving the diagnosis, monitoring, and personalization for the treatment of these conditions. By elucidating how cannabinoids modulate EV biogenesis, cargo, and signaling, this framework may uncover key molecular and cellular mechanisms underlying neuropsychiatric dysfunctions, thereby helping to overcome the persistent limitations of current pharmacological interventions and ultimately improve clinical endpoints by promoting treatment adherence via personalized medicine.

## 2. Endocannabinoids and Neurological and Psychiatric Disorders

Considerable progress has been made over the last century in understanding Cannabis sativa’s active compounds as well as relevant pharmacological targets. Roger Adams isolated the major non-psychotomimetic molecule, cannabidiol (CBD), in the early 1900s [17], followed by the identification of Δ9-tetrahydrocannabinol (THC) by Mechoulam and Gaoni [18]. These findings stimulated the research into their molecular targets, ultimately leading to the discovery of CB1 and CB2 cannabinoid receptors, their endogenous ligands, including anandamide (AEA) and 2-arachidonoyl-glycerol (2-AG), as well as the enzymes responsible for their synthesis and degradation [19,20,21,22].

Since then, growing evidence has identified novel compounds with binding properties not only to classical cannabinoid receptors 1 and 2 (CB1 and CB2), but also to newly characterized molecular targets. Consequently, the concept of the endocannabinoid system has evolved into a broader framework, often referred to as the “expanded endocannabinoid system” or “endocannabinoidome,” encompassing a complex network of lipid mediators, receptors, and metabolic pathways [23].

The endocannabinoid system (ECS) is one of the most pleiotropic signaling systems in vertebrates and plays a crucial role in maintaining central nervous system homeostasis. Classical endocannabinoids, primarily AEA and 2-AG, are synthesized on demand in postsynaptic neurons and act retrogradely to modulate presynaptic neurotransmitter release [24]. Once endocannabinoids are produced in the postsynaptic neuron, they diffuse across the synaptic cleft and bind to cannabinoid receptors, mainly CB1, located on presynaptic terminals. Activation of CB1 receptors leads to the inhibition of neurotransmitter release, including glutamate and GABA (γ-aminobutyric acid), thereby modulating both excitatory and inhibitory transmission. This retrograde signaling enables the ECS to fine-tune excitatory and inhibitory synaptic transmission, thus maintaining synaptic balance and preventing excessive neuronal activity [25,26].

Indeed, depending on whether CB1 receptors are expressed at excitatory or inhibitory synapses, their retrograde activation by endocannabinoid release can induce both short-term and long-term forms of synaptic plasticity, such as long-term depression (LTD) and long-term potentiation (LTP) [27,28,29,30]. Through this mechanism, endocannabinoid signaling contributes to the fine-tuning of neuroplastic processes that are fundamental for cognitive functions, including learning, memory, and emotional regulation.

Besides synaptic modulation, the ECS plays a key role in regulating neuroinflammatory processes within the central nervous system. For instance, CB2 receptors, which are primarily expressed on microglial cells, can suppress the release of pro-inflammatory cytokines upon activation by endocannabinoids, thereby mitigating neural tissue damage [31,32,33,34]. This mechanism is particularly relevant under pathological conditions such as neurodegeneration, traumatic brain injury, or viral infection, in which microglial activation contributes to secondary neuronal injury.

Despite being markedly expressed in neurons, CB1 receptors are also present in astrocytes and regulate neuron-glia communication and the release of gliotransmitters. Indeed, ECS signaling in astrocytes has been implicated in the homeostasis of calcium dynamics, which are essential for maintaining extracellular neurotransmitter balance and preventing excitotoxicity [35,36,37].

Beyond the canonical ECS, recent research has led to the recognition of the so-called “endocannabinoidome”, or “expanded endocannabinoid system”, which encompasses not only the CB1 and CB2 receptors, but also a variety of enzymes, lipid mediators, and non-cannabinoid receptors, such as TRPV1, PPARs, and GPR55 [38]. This extended concept emphasizes the complicated interplay between endocannabinoid signaling and other neurotransmitter and neuromodulatory systems, implying a more general regulatory role in maintaining homeostasis across numerous physiological domains. The endocannabinoidome thus offers a broader viewpoint for identifying novel pharmaceutical targets outside of the classic ECS.

Dysregulation of the ECS has been increasingly associated with the pathophysiology of several neurological and psychiatric disorders. Alterations in the expression or function of cannabinoid receptors, in the levels of endocannabinoids, and in the activity of ECS-related enzymes have been documented in both human patients and animal models of various neuropsychiatric conditions [38,39]. Given that the ECS closely interacts with other major neurotransmitter systems, such as glutamatergic, dopaminergic, and serotonergic pathways, these maladaptive changes can disrupt key neural processes, including synaptic plasticity, emotional regulation, neuroinflammation, and stress reactivity. Ultimately, such dysregulation contributes to the onset and progression of these disorders [40,41,42,43,44,45,46,47,48,49].

Importantly, the role of the endocannabinoidome in psychiatric and neurological disorders highlights its therapeutic potential. Anxiety, depression, schizophrenia, and neurological disorders have all been linked to dysregulation in this network, as altered lipid mediator profiles and receptor signaling pathways lead to pathological states [50,51,52]. Targeting endocannabinoidome components may help to restore synaptic homeostasis, control inflammatory responses, and enhance neuroprotection.

Altogether, the ECS emerges as a multifaceted neuromodulation, and growing evidence supports its involvement in a wide spectrum of physiological and pathological brain processes. Understanding its complex interactions offers promising avenues for the development of novel therapeutic strategies targeting both neurological and psychiatric disorders.

### 2.1. Cannabinoids and Neurodegenerative Disorders

Neurodegenerative disorders, such as Alzheimer’s, Parkinson’s, Huntington’s disease, and amyotrophic lateral sclerosis, are characterized by neuronal dysfunction and loss, which is often associated with neuroinflammation, increased oxidative stress, and excitotoxicity [53]. Despite progress in symptomatic treatments, disease-modifying medications are still restricted, emphasizing the critical need for innovative therapeutic methods. Cannabinoids have emerged as interesting possibilities in this context due to their pleiotropic mechanisms of action, which go beyond classical neurotransmission to modulate immunological responses, mitochondrial function, and synaptic plasticity.

Extensive preclinical research has shown that modulating the ECS has consistent neuroprotective effects across a variety of neurodegenerative disease models. For instance, cannabinoids effectively reduce Aβ accumulation and tau pathology in Alzheimer’s disease (AD) models. CBD has been shown in transgenic mouse models to attenuate Aβ-induced neuroinflammation and prevent memory impairments by activating PPARγ pathways [54,55,56]. We also have evidence that inhibiting monoacylglycerol lipase (MAGL), which degrades 2-arachidonoylglycerol, significantly reduces β-amyloid-induced neurodegeneration via CB1 receptor-dependent processes [57].

Moreover, this effect has also been observed with selective CB1 and CB2 receptor agonists and mixed CB1/CB2 agonists. For instance, although the selective CB1 receptor agonist arachidonyl-2-chloroethylamide (ACEA) does not modulate production, aggregation, and clearance of Aβ, it mitigates the cytotoxic effects of Aβ-42 oligomers in primary cortical neuron cultures [58]. This leads to a reduction in astroglial reactivity around Aβ plaques and lower expression of pro-inflammatory markers in AD transgenic mice.

Additionally, chronic treatment with JWH-133 and WIN55,212-2, a CB2 selective agonist and mixed CB1/CB2 agonist, respectively, led to enhanced cognitive performance in two distinct transgenic mouse models of brain amyloidosis. The effectiveness of cannabinoid compounds in mitigating cognitive deficits was inversely correlated with the stage of disease progression at the onset of treatment, with greater benefits observed when administered during earlier phases [55,59].

Regarding the impact of cannabinoids on motor function, findings from animal models of Parkinson’s disease have been heterogeneous and at times contradictory. Activation of CB1 receptors has been associated with reductions in akinesia, motor deficits, and tremor, possibly through mechanisms not directly dependent on receptor activity [60]. In contrast, another study indicated that administration of Δ9-THC or the cannabinoid agonist levonantradol in primates led to an aggravation of bradykinesia, while treatment with a CB1 receptor antagonist did not mitigate MPTP-induced Parkinsonian symptoms [61].

Moreover, CB2 receptor agonists like JW015 and AM1241 showed considerable neuroprotection in MPTP-induced toxicity tests, preserving dopaminergic neurons and enhancing motor function [62]. Also, the neuroprotective compound CP55,940 demonstrated effectiveness against paraquat-induced Parkinsonism via antioxidant pathways unrelated to cannabinoid receptors [63].

Clinical evidence for endocannabinoid-based interventions in neurodegenerative diseases remains limited but increasingly promising. A comprehensive systematic review and meta-analysis examining the endocannabinoid system in Alzheimer’s disease revealed significant alterations in ECS components, with AD patients showing decreased CB1 receptor expression, particularly in the frontal cortex (SMD = −1.09, *p* < 0.01), and increased 2-arachidonoylglycerol levels (SMD = 0.46, *p* = 0.02) [64].

Interestingly, Bahji and colleagues (2019) conducted a comprehensive review and meta-analysis that found preliminary but consistent data supporting cannabinoids’ efficacy and tolerability for treating neuropsychiatric symptoms (NPS) in dementia [65]. The authors discovered that cannabinoids significantly reduced agitation, aggression, and nocturnal disturbances, as measured by standardized tools such as the Cohen-Mansfield Agitation Inventory and the Neuropsychiatric Inventory, after analyzing nine clinical trials with 205 participants, the majority of whom were Alzheimer’s disease patients.

Regarding clinical evidence on the effects of cannabinoids in Parkinson’s disease, an open-label trial administered CBD for four weeks to six patients presenting with positive symptoms, such as illusions, delusions, and hallucinations, as well as negative symptoms, including social withdrawal and depression. The treatment was associated with a significant reduction in psychotic manifestations, as measured by the Parkinson Psychosis Questionnaire (PPQ) and the Brief Psychiatric Rating Scale (BPRS) [66].

Furthermore, in a randomized trial, 21 patients with Parkinson’s disease were allocated to receive either a placebo or daily CBD at doses of 75 mg or 300 mg, with seven participants per group. The compound, prepared in powdered form, dissolved in corn oil and encapsulated in gelatin, was administered for six weeks. Outcomes were evaluated using the UPDRS motor score and the PDQ-39 quality-of-life scale. While no difference between groups was observed in motor function, patients receiving 300 mg/day of CBD showed a significant improvement in overall quality of life compared with baseline and placebo [67].

Importantly, cannabinoids were generally well tolerated, with 93% of trials completed and no serious treatment-related side events observed. These findings suggest that cannabis may represent a safer pharmacological option to standard treatments such as antipsychotics. However, it is important to stress that the overall quality of evidence was low, and bigger, well-designed trials are still necessary to prove their therapeutic relevance.

### 2.2. Cannabinoids and Neurodevelopmental Disorders

Neurodevelopmental disorders comprise a heterogeneous group of conditions characterized by alterations in brain development that manifest as neuropsychiatric symptoms. As extensively discussed by Thapar et al. [68], the concept of neurodevelopmental disorder can be broadly used to group a wide range of conditions that exhibit clinically distinct neurological and psychiatric symptoms, such as autism spectrum disorder (ASD), schizophrenia, and attention-deficit hyperactivity disorder (ADHD). Despite their etiological diversity, one key defining characteristic is the early onset of symptoms, typically during childhood or puberty, which reflects the underlying disruption in neurodevelopmental trajectories.

Emerging evidence suggests that ECS dysregulation contributes to the pathogenesis of several neurodevelopmental disorders. For instance, altered levels of endocannabinoids, changes in CB1 and CB2 receptor expression, and disruption of crosstalk between the ECS and other neurotransmitter systems have been described in schizophrenia and autism [69,70,71]. Given the ECS’s central role in neuroplasticity, neuroinflammation, and cognitive/emotional regulation, these findings promote a rationale for exploring cannabinoids as useful therapies in the context of neurodevelopmental disorders.

As discussed earlier, animal models reveal that CBD exhibits neuroprotective, anxiolytic, and anti-inflammatory effects, modulating receptor activity such as CB1, CB2, and serotonin 5HT1A receptors [72]. These data suggest that cannabinoid treatment during critical developmental windows may mitigate or prevent behavioral and cognitive deficits associated with neurodevelopmental disorders like ASD and schizophrenia. Furthermore, some preclinical findings indicate that early CBD administration can produce long-lasting improvements in neural function and behavior beyond the treatment period, supporting its potential as a disease-modifying therapy [73].

Clinical evidence examining cannabinoid use in neurodevelopmental disorders remains limited but promising. Most clinical trials focus on pediatric populations with ASD, ADHD, and other developmental disabilities. Studies have reported improvements in core symptoms such as anxiety, irritability, sleep disturbances, and social communication following CBD treatment [73]. Importantly, CBD appears to have a favorable safety profile in these groups, with mild side effects such as somnolence and appetite changes reported. However, rigorous randomized controlled trials (RCTs) are scarce, and many existing studies are observational or open studies, necessitating cautious interpretation of efficacy outcomes.

Despite emerging positive signals, the clinical evidence base is hindered by methodological challenges, including small sample sizes, heterogeneous study designs, and variable cannabinoid formulations. Consequently, there is insufficient high-quality data to definitively recommend cannabinoids as standard treatments for neurodevelopmental disorders. Researchers advocate for larger, well-controlled RCTs to clarify optimal dosing, long-term safety, and differential responses across neurodevelopmental conditions. Alongside clinical research, mechanistic investigations continue to elucidate how cannabinoids may influence neuroinflammatory pathways and neuroplasticity, aiming to optimize therapeutic windows and outcomes.

In summary, both preclinical and clinical evidence imply that cannabinoids, notably CBD, hold therapeutic potential for managing symptoms and potentially modifying the course of neurodevelopmental disorders. Preclinical models provide compelling support for early intervention during critical developmental periods, while clinical studies, though few, suggest symptom improvements with acceptable safety. Continued research is essential to establish evidence-based guidelines for cannabinoid use in this vulnerable population, balancing potential benefits against risks to inform clinical practice.

### 2.3. Cannabinoids and Psychiatric Disorders

Psychiatric disorders can be characterized as conditions marked by clinically relevant alterations in cognition, emotional regulation, or behavior arising from dysfunctions in psychological, biological, or developmental mechanisms that support mental functioning [74]. Despite being historically separated from neurological problems, mental illnesses are now more often recognized as brain-based conditions that result from intricate relationships between developmental paths, environmental exposures, and hereditary susceptibility. Their clinical variability and the difficulties in accurately diagnosing and treating them are exacerbated by their complex nature [75].

The diagnosis of psychiatric disorders relies exclusively on clinical symptoms evaluated through standardized classificatory systems such as the Diagnostic and Statistical Manual of Mental Disorders (DSM-V) [76] and the International Classification of Diseases (ICD-11) [77]. Instead of using objective biological indicators, these frameworks describe disorders based on patterns of functional impairment and clusters of symptoms, reflecting current clinical consensus. Although crucial for clinical practice, this symptom-based approach highlights the need for biomarkers and mechanistic insights by introducing heterogeneity and overlap between conditions.

Preclinical research demonstrates that inhibiting FAAH with compounds such as URB597 and PF-3845 produces robust anxiolytic and antidepressant-like effects in rodent models of chronic stress, social defeat, and anxiety [78,79,80], and newer FAAH inhibitors such as JNJ-42165279 have shown promising clinical results, including increased AEA levels and reduced activation of anxiety-related brain regions in healthy volunteers [81] as well as improved symptoms in individuals with social anxiety disorder [82]. Inhibition of MAGL, the primary enzyme responsible for 2-AG degradation, also represents a valuable strategy: elevated MAGL expression is associated with heightened anxiety-like behaviors, whereas MAGL blockade enhances 2-AG signaling and reduces vulnerability to stress-induced anxiety [83,84,85], with inhibitors such as JZL184, KML29, and MJN110 consistently decreasing anxiety-like behaviors across multiple rodent paradigms [86,87,88]. In contrast to these indirect modulators, direct agonists or antagonists of CB1 and CB2 receptors—including phytocannabinoids from Cannabis sativa and synthetic analogs—exert more immediate effects on the endocannabinoid system, broadening the pharmacological landscape for targeting anxiety and related disorders.

In the case of the two major compounds found in Cannabis sativa, THC and CBD, most studies consistently report anxiogenic effects following THC administration. Human studies consistently show that THC produces dose-dependent anxiogenic effects, as demonstrated in a pooled analysis of four double-blind, placebo-controlled trials using vaporized or oral THC at low (5–10 mg) and high (20–25 mg) doses in healthy adults with no cannabis history [89], as well as in a separate study where intravenous THC (0.015–0.03 mg/kg) similarly increased self-reported anxiety in a dose-dependent manner in largely cannabis-naïve participants [90]. In contrast, preclinical findings are more mixed: intraperitoneal THC administered across a wide dose range (0.2–6.4 mg/kg) produced anxiolytic-like effects at higher doses in the elevated plus maze without altering locomotion in the open field test [91], though these results were obtained in non-stressed animals, limiting conclusions about THC’s potential anxiolytic properties under pathological or stress-related conditions.

In contrast to THC, CBD does not produce psychotomimetic effects and exhibits low affinity for endogenous cannabinoid receptors. Instead, CBD is hypothesized to exert modulatory effects across multiple systems in the brain and peripheral tissues, influencing not only the endocannabinoid system but also serotonergic and adenosine-mediated neurotransmission [92].

Preclinical research consistently shows that CBD exerts anxiolytic and antidepressant effects across diverse animal models. Early studies using the elevated plus maze demonstrated an inverted U-shaped dose–response curve, with moderate CBD doses producing anxiolytic-like effects [93,94]. These results were later confirmed using the Vogel conflict test, contextual fear conditioning, and a murine PTSD model, where CBD (10 mg/kg) induced anticonflict behaviors comparable to benzodiazepines, reduced freezing behavior, and attenuated predator-induced anxiety [95,96,97]. CBD has also shown antidepressant-like effects in the forced swim and tail suspension tests, reducing immobility similarly to imipramine and reversing stress-induced anhedonia and reduced hippocampal neurogenesis in a chronic unpredictable stress model—effects mediated through 5-HT1A receptor activation [98,99].

Clinical evidence similarly supports the anxiolytic properties of CBD, particularly in social anxiety disorder (SAD). In a double-blind study, Bergamaschi et al. [100] reported that 600 mg of oral CBD significantly reduced anxiety, cognitive impairment, and physiological stress during a simulated public speaking test. A second placebo-controlled study testing 100, 300, and 900 mg of CBD found that only the 300 mg dose produced anxiolytic effects, consistent with a U-shaped dose response [101]. Additional research shows that CBD can reduce cue-induced anxiety and craving in individuals with heroin use disorder, with effects lasting up to one week after the final dose [102].

Despite strong pre-clinical support, the antidepressant effects of CBD remain insufficiently explored clinically. Indirect evidence comes from a prospective open-label trial showing that CBD reduced burnout and emotional exhaustion in frontline healthcare workers during the COVID-19 pandemic [103]. However, clinical studies specifically targeting depressive disorders are lacking, underscoring the need for well-designed, large-scale trials to determine CBD’s efficacy and safety as a treatment for depression. Preclinical and early clinical findings suggest that cannabinoids—particularly CBD—and enzymatic inhibitors of the endocannabinoid system, such as FAAH and MAGL inhibitors, hold significant therapeutic potential for anxiety and depressive disorders. However, the complexity of endocannabinoid signaling and the variability in clinical outcomes highlight the need for more rigorous, large-scale trials to clarify efficacy, safety, and optimal dosing. Ultimately, the successful clinical integration of these approaches will depend on resolving current knowledge gaps and tailoring treatments to patient-specific needs.

## 3. Biomarkers in the Context of Neuropsychiatric Disorders

The identification of biomarkers has transformed many areas of modern medicine. Biomarkers—defined as endogenous molecules that provide measurable indicators of physiological or pathological states—are essential for diagnosis, monitoring therapeutic responses, and predicting disease risk or progression [104]. Two main strategies guide biomarker discovery: (1) hypothesis-driven approaches that target molecules based on known disease mechanisms, and (2) unbiased approaches using high-throughput molecular technologies (e.g., multi-omics) to distinguish healthy individuals from patients [105]. Although remarkable advances have been made, the translation of biomarkers into clinical practice remains challenging. While examples such as prostate-specific antigen (PSA) illustrate how a single biomarker can revolutionize clinical care [106], many proposed biomarkers have failed to achieve clinical utility, demonstrating that individualized, blood-based precision medicine is still aspirational in most fields.

Peripheral blood markers are attractive due to their accessibility and the systemic nature of circulating molecules. However, for conditions centered in the central nervous system, the blood–brain barrier (BBB) complicates interpretation, raising legitimate concerns about how accurately peripheral markers reflect central processes [107]. Despite this, advances in detection technologies, integrative analytics, and multimodal approaches—particularly the combination of blood-based markers with neuroimaging techniques such as fMRI and PET—are increasingly refining biomarker signatures for neurological and psychiatric conditions [108]. In AD, for example, these multimodal strategies have already begun to reshape diagnostic pathways and patient follow-up, with the recent introduction of ultrasensitive blood tests aiming at measuring levels of amyloid β peptides, tau, glial fibrillar acidic protein (GFAP), and neurofilament light (NfL) protein 2 [109].

In neuropsychiatric disorders, the search for reliable biomarkers is still at an early stage. Although numerous genetic loci, genes, and single-nucleotide polymorphisms have been associated with vulnerability to psychiatric and neurodegenerative conditions, few findings have translated into clinically actionable biomarkers. Current candidates include microRNAs (miRNAs), brain-derived neurotrophic factor (BDNF), dopaminergic receptor subunits, DNA methylation patterns, inflammatory cytokines (e.g., IL-6, TNF-α, IL-1β), complement system components [110], proteins involved in mitochondrial function, and neurotransmitters such as serotonin, GABA, dopamine, and glutamate [111,112].

Genetic changes in the endocannabinoid system genes (ECS) and circulating levels of ECBs have been proposed as potential biomarkers for neuropsychiatric disorders. Clinical and translational studies consistently demonstrate that circulating levels of anandamide (AEA) and 2-arachidonoylglycerol (2-AG) reflect disease states and symptom severity across several conditions. For example, antipsychotic-naïve or early-stage schizophrenia patients show elevated AEA levels in cerebrospinal fluid, which inversely correlate with psychotic symptom severity [113,114,115]. A systematic review found that patients with psychosis exhibit a higher ECS tone (elevated AEA in CSF and blood, increased CB1 expression) that normalizes after treatment [116]. In stress-related disorders such as PTSD, greater circulating AEA has been associated with lower mood disturbance and symptom severity, and its mobilization in response to acute stress (e.g., exercise) is altered in patients [117]. Beyond lipid ligands, changes in ECS-related enzymes (e.g., FAAH) and receptors (CB1/CB2) have also been documented in neuropsychiatric populations, underscoring their potential as molecular biomarkers of disease state and treatment response [39].

However, most of these markers lack specificity, likely reflecting the substantial biological heterogeneity that characterizes psychiatric conditions and the inherent individuality of human physiological responses. Thus, many of these candidates ultimately serve better as state markers, indicating the presence or severity of a condition—rather than causal or diagnostic markers. A further challenge emerges from the fact that neurotransmitters, neurotrophic factors, and related receptor systems have distinct roles in peripheral tissues, raising important questions: Can peripheral biomarkers truly reflect brain physiology? Do they represent signals originating from the CNS, peripheral processes that secondarily influence the brain, or a bidirectional communication loop? Distinguishing these sources is crucial for clinical translation.

Fortunately, emerging work on EVs, including those derived specifically from neural cells, offers a promising solution. EVs can cross the BBB bidirectionally and carry molecular cargo (proteins, lipids, RNA species) that reflects the physiological state of their cells of origin, providing a minimally invasive window into CNS processes. As technologies to isolate and characterize CNS-derived EVs continue to advance, they are increasingly recognized as one of the most promising avenues for developing brain-relevant biomarkers in psychiatry and neurology.

## 4. Extracellular Vesicles: Definitions and Characteristics

EVs are nano-scale particles encased in a lipid bilayer that are released by diverse cells into the extracellular environment. Exosomes, microvesicles, and apoptotic bodies are three types of vesicles that play an important role in cellular communication (Figure 1). Exosomes, with a diameter of 30–150 nm, are released by the fusion of multivesicular bodies (MVBs) with the plasma membrane, whereas microvesicles, which are bigger (100–1000 nm), are directly generated by the fragmentation of the plasma membrane [118].

Although EVs were initially reported in the 1940s, their biological and physiological relevance was not well recognized until recently for a detailed review, see Couch et al. [119]. Their research gained significant momentum with advances in molecular biology and electron microscopy, which allowed for the precise identification of these vesicles and an understanding of their essential roles in processes such as cellular homeostasis, immune responses, and even the development of neurodegenerative diseases. Furthermore, increasing interest in EVs is associated with their capacity to transport proteins, lipids, RNA, and even DNA, which allows cells to communicate critical information about their physiological and pathological states and modulate immune responses [120].

In the context of the CNS, EVs are released by neurons, astrocytes, oligodendrocytes and microglial cells into the interstitial fluid of the brain and spinal cord parenchyma [121]. They function as vehicles for the transfer of complex signals between the neural and glial cells, facilitating omnidirectional communication among diverse cell populations. Under physiological conditions, EVs contribute significantly to CNS homeostasis through mechanisms that include waste clearance, trophic support of neurons, antigen presentation, and maintenance of myelin and synaptic plasticity [122,123,124]. However, we are only beginning to unravel their role in pathological conditions such as those related to neuropsychiatric disorders (Table 1).

### 4.1. Extracellular Vesicles in Neurodegenerative Diseases

In neurodegenerative diseases, both the quantity and molecular composition of EVs are often altered, suggesting their involvement in disease onset and progression. EVs play a dual role in CNS disorders: on one hand, cells use them to remove toxic proteins and aggregates from their cytoplasm; on the other hand, these same nanoparticles can interact with healthy cells, delivering their toxic cargoes and potentially spreading disease [137,138]. Indeed, this has been observed in several conditions including Alzheimer’s disease (AD), Parkinson’s disease (PD), amyotrophic lateral sclerosis, and Huntington’s disease, where EVs have been implicated as carriers of misfolded proteins associated with these disorders.

In Alzheimer’s disease, research has demonstrated that microglial Aβ-EVs significantly affect synaptic plasticity both in cultured neurons and in brain slices. When injected into mouse brains, these EVs propagate synaptic dysfunction in the entorhinal-hippocampal circuit through a mechanism sensitive to annexin-V, a phosphatidylserine ligand that blocks EV extracellular motion [125]. Indeed, it was shown that Aβ exposed on EV surfaces accounts for synaptic dysfunction, as only EVs carrying the peptide decrease dendritic spine size and impair synaptic plasticity both in vitro and ex vivo.

Noteworthy, the action of Aβ-EVs perfectly mimics that of soluble oligomeric Aβ42, which impairs long-term potentiation in brain slices by acting primarily on the post-synaptic site. Even more significantly, the concentration of EV surface Aβ42 that impairs LTP is considerably lower than that of oligomeric Aβ42 alone, suggesting that EVs enhance the synaptotoxicity of Aβ on neuronal circuitry.

In PD, EVs derived from neurons, microglia, and red blood cells carry pathogenic proteins such as α-synuclein (aSyn) and leucine-rich repeat kinase 2 (LRRK2), which are significantly elevated in patient-derived EVs compared to controls [139]. Interestingly, neuronal L1CAM-positive blood EVs show increased aSyn levels, while red blood cell-derived EVs (RBC-EVs) demonstrate elevated aSyn concentrations that correlate with motor symptom severity on the MDS-UPDRS scale.

Beyond proteins, there are also changes in microRNA profiles in extracellular vesicles from patients with Parkinson’s disease. Studies identified diagnostic signatures-such as miR-126-5p, miR-99a-5p, and miR-501-3p in cerebrospinal fluid EVs-that distinguish PD patients from healthy individuals and those with prodromal REM sleep behavior disorder (iRBD) [127]. Critically, microglial EVs contribute to disease progression by facilitating the intercellular transmission of pathological aSyn aggregates, which induce neuronal protein aggregation and neurodegeneration, as demonstrated in both in vitro and in vivo models [140].

### 4.2. Extracellular Vesicles in Neurodevelopmental Disorders

In schizophrenia, research has uncovered EV-based complement system dysregulation as a novel disease mechanism. Proteomic profiling of plasma EVs identified elevated C1q, C3, and C5b-9 levels (e.g., components of the classical and terminal complement pathways) that distinguished schizophrenia patients from healthy controls with 89.5% accuracy [141]. Machine learning models integrating these complement proteins achieved superior diagnostic performance (AUC = 0.966) in differentiating schizophrenia from bipolar disorder, highlighting pathway-specific alterations.

Besides that, several studies have shown differences in the miRNA extracellular vesicle’s cargo between EVs isolated from healthy and schizophrenic patients. For instance, a study has found that patients with schizophrenia had considerably higher levels of miR-497 in their prefrontal cortex-derived extracellular vesicles compared to healthy donors [129]. This miRNA has been reported to negatively regulate BCL-2-related pathways, along with the enhancement of neurodegenerative and oxidative stress processes [142].

Along with changes in the microRNA content, we also have evidence that points to differences in the protein content of extracellular vesicles isolated from schizophrenic patients with schizophrenia. A study found 52 differentially expressed proteins (DEPs) in schizophrenia (SCZ) patients compared to healthy controls (HC), and 18 DEPs in antipsychotic responders (RES) versus non-responders (NRES) [130]. Five proteins were revealed to be upregulated in SCZ: FN1, ITGB3, PECAM1, ITGA6, and CD5. In addition, when RES was compared to NRES, six SCZ-associated proteins displayed differential expression, including overexpression of CD33, CD40, CD36, TENM2, and EGFR, as well as downregulation of ITGB3.

Moreover, another study revealed elevated levels of GFAP and Aβ42, together with reduced α-II-spectrin in EVs isolated from schizophrenic patients [131]. The increase in GFAP, a well-known marker of astrocytic activation and neuroinflammation, supports the idea that immune system dysregulation contributes to schizophrenia pathophysiology. Decreased α-II-spectrin, a key cytoskeletal protein for neuronal integrity, indicates continuous neuronal injury and is positively connected with psychopathology severity. Elevated Aβ42 levels suggest common pathways with neurodegenerative diseases, since they increase neuroinflammation via NF-κB signaling and degrade neuronal survival through mitochondrial malfunction.

These alterations highlight the interplay between immune dysregulation, extracellular matrix remodeling, and vascular dysfunction in the pathophysiology of schizophrenia. Furthermore, it also enhances our comprehension about the concept that schizophrenia is not solely a disorder of impaired neurotransmission, but rather a complex interaction between neuroplastic and neuroimmune alterations.

Importantly, the role of EVs as conveyors of disease-relevant signals in neurodevelopmental disorders also extends to ASD. EVs derived from ASD patients have been found to carry altered protein and microRNA cargos that influence neural connections, synaptic plasticity, and immunological modulation.

Recent proteome profiling of plasma-derived EVs found a selective downregulation of five proteins associated with ASD (WWP2, HSP27, CLEC1B, CD40, and FRα). These proteins are involved in EV synthesis, cellular stress responses, immunological signaling, and folate metabolism [132]. Unlike the broad alterations often observed in proteomic studies, this restricted set of changes suggests a more specific molecular signature of ASD. Notably, reduced HSP27 and CD40 reinforce the contribution of impaired stress resilience and neuroinflammation, while downregulated FRα supports long-standing evidence of disrupted folate pathways in autism.

Furthermore, another study revealed that EVs isolated from children diagnosed with ASD significantly stimulate cultured human microglial cells to secrete pro-inflammatory markers such as IL-1β, when compared to controls [133]. Although with important limitations, this study highlights the interplay between extracellular vesicles, neuroimmune dysregulation and ASD pathophysiology.

Collectively, these findings strengthen the notion that EVs are not merely byproducts of cellular activity but active participants in ASD pathophysiology, holding promise as non-invasive biomarkers for early detection and intervention.

### 4.3. Extracellular Vesicles in Psychiatric Disorders

These dynamic changes in EVs content are also important in the context of psychiatric disorders such as major depression (MDD) and anxiety disorders. For instance, in MDD, plasma-derived EVs exhibit distinct miRNA profiles that regulate hypothalamic–pituitary–adrenal (HPA) axis activity and hippocampal neurogenesis. A landmark study demonstrated that intravenous injection of MDD patient-derived EVs into mice induced depressive-like behaviors, an effect mediated by elevated miR-139-5p levels [134]. The authors have shown that miR-139-5p suppresses neural stem cell proliferation and neuronal differentiation by targeting Wnt/β-catenin and BDNF signaling pathways, directly linking EV cargo to impaired neurogenesis.

Lastly, mice subjected to restraint stress, a recognized model of stress-related anxiety behavior, brain-derived extracellular vesicles enriched with miR-199a-3p, miR-99b-3p, and miR-140-5p reduced anxiety-like behaviors in the elevated plus compared to controls [135]. Interestingly, the authors discuss that miR-199a-3p suppresses Mecp2 translation in hippocampal neurons, enhancing BDNF expression and the genesis of dendritic spines.

Taken together, these findings underscore the multifaceted roles of extracellular vesicles as both mediators and mirrors of neuropathological processes in the central nervous system. By shuttling diverse molecular cargo-including proteins, lipids, and regulatory RNAs-EVs not only facilitate intercellular communication under physiological conditions but also contribute to the propagation and modulation of disease states in neurodegenerative and psychiatric disorders.

Despite these advances, many aspects of EV biology in the CNS remain to be elucidated, particularly regarding how external modulators may influence their function and content. Notably, recent research has begun to explore the intriguing crosstalk between cannabinoids and EVs, suggesting that the endocannabinoid system may regulate, or be regulated by, EV-mediated signaling in the brain. Understanding this interface could reveal novel mechanisms underlying neuroprotection, neuroinflammation, and synaptic plasticity, paving the way for new diagnostic and therapeutic strategies. The following section will underscore the current knowledge and emerging questions at the intersection of cannabinoids and extracellular vesicle biology.

## 5. Cannabinoids-Extracellular Vesicles Interface

A paucity of evidence has now supported the functional relevance of EVs in intercellular communication and their role in endocannabinoid signaling (Figure 2).

Gabrielli et al. [143] presented the first direct proof that microglia-derived EVs contain the endocannabinoid anandamide (AEA) on their surface, supporting the idea that EVs operate as shuttles for bioactive lipids and neuroactive chemicals inside the central nervous system. Through a series of electrophysiological, biochemical, and imaging experiments, the authors demonstrated that microglial EVs, particularly microvesicles (MVs), are enriched in AEA and capable of modulating inhibitory neurotransmission by activating CB1 receptors on GABAergic neurons. Application of microglia-derived MVs to primary hippocampal cultures resulted in a significant decrease in the frequency of miniature inhibitory postsynaptic currents (mIPSCs), an effect that was abolished by the CB1 antagonist SR141716A.

This inhibitory modulation was not due to endocytosis or the release of soluble factors, as the vesicular structure and surface localization of AEA were shown to be essential for biological activity. Moreover, the study confirmed activation of downstream CB1 signaling by showing ERK phosphorylation upon exposure to AEA-enriched MVs. These results suggest that microglia not only synthesizes and release endocannabinoids but also uses EVs as platforms for targeted signaling to neurons.

Lombardi and colleagues [144] also demonstrated that immune phagocytic cells, like macrophages and microglial cells, can release EVs containing both AEA and 2-AG to induce myelination via oligodendrocyte proliferation in vitro. In addition, the authors also demonstrate that the circulating peripheral monocytes can also release EVs enriched with endocannabinoids.

Conversely, more recently, Straub et al. [145] suggested that in a coculture system using a reporter cell line engineered to express the fluorescent endocannabinoid sensor GRABeCB2.0, neurons can secrete extracellular vesicles enriched with 2-AG—but not anandamide. This release occurs in a stimulus-dependent manner and is controlled by signaling pathways involving protein kinase C, diacylglycerol lipase, and the small GTPase Arf6. Moreover, the process is disrupted by compounds that block facilitated diffusion of endocannabinoids.

This discovery closes a long-standing gap in our understanding of how extremely lipophilic endocannabinoids are produced on demand and travel across the extracellular space to reach cannabinoid receptors. Their findings establish EVs as a physiologically relevant transport for AEA, coupling vesicle-mediated intercellular signaling to synaptic activity regulation via the endocannabinoid system.

Building upon this data, another study has demonstrated a dynamic and complex relationship between the endocannabinoid system and the release of extracellular vesicles. Brandes et al. [146] demonstrated that, in patients with COVID-19, the physical association of multiple endocannabinoids, including 2-arachidonoylglycerol (2-AG), palmitoylethanolamide (PEA), oleoylethanolamide (OEA), and stearoylethanolamide (SEA), EVs isolated from plasma points to a selective vesicular packaging of lipid mediators. Especially, anandamide (AEA) did not co-localize with EVs, confirming earlier results on viable alternative routes of transport.

### 5.1. Possible Mechanisms Involved in the Interface Between EV and Cannabinoids

The mechanisms underlying the interaction between cannabinoids and EVs remain largely unexplored, as only a limited number of studies have directly addressed this scientific question. Nevertheless, several cellular processes offer plausible mechanistic hypotheses worth investigating.

Cannabinoid receptors, particularly CB1, exhibit high constitutive activity [147]. Under basal conditions, a significant proportion of CB1 receptors resides intracellularly, especially within early endosomes [148]. Upon agonist binding, CB1 undergoes endocytosis and traffics through early and sorting endosomes (SE), eventually being recycled back to the plasma membrane through Rab GTPase–dependent pathways. Rab GTPases are essential regulators of membrane trafficking and EV biogenesis [149], coordinating vesicle budding, transport, docking, and fusion events [150]. Because EVs are generated through the fusion of late endosomes/multivesicular bodies (MVBs) with the plasma membrane, Rab-dependent transitions between early and late endosomes could represent a mechanistic link between cannabinoid receptor signaling and EV release.

EV secretion is also implicated in the disposal of unwanted cellular components, including dysfunctional mitochondria. This phenomenon was first described during reticulocyte maturation [151]. More recently, in the aging mouse heart, deletion of a small GTPase isoform increased EVs containing mitochondrial material, suggesting that mitochondrial stress or impaired lysosomal activity may activate EV-mediated clearance pathways [152].

Cannabinoid receptors themselves are expressed in mitochondria, where mitochondrial CB1 (mtCB1) signaling reduces cAMP production and regulates intracellular calcium, bioenergetics, apoptosis, mitochondrial dynamics, mitophagy, and biogenesis [153,154]. Deleting mtCB1 alters cannabinoid effects on hippocampal memory processing, and CBD has been shown to decrease mitochondrial respiration in trophoblasts [155]. These findings raise the hypothesis that cannabinoid-mediated changes in mitochondrial homeostasis might influence EV-mediated mitochondrial disposal pathways.

Additionally, CB1 receptors appear to be localized to lysosomal membranes. In a model of amyloid-β–induced lysosomal destabilization, endocannabinoid-mediated CB1 activation restored lysosomal function and exerted neuroprotective effects [156]. Because lysosomal dysfunction is tightly linked to altered EV secretion, this represents another potential interface between ECS signaling and EV biology.

A further mechanistic intersection involves ceramide. Ceramides are key sphingolipid components of EV membranes and essential for EV biogenesis and release [157]. Cannabinoid receptor activation promotes ceramide formation through sphingomyelin hydrolysis or de novo synthesis [158], suggesting that cannabinoids may regulate EV production via ceramide-dependent pathways.

Taken together, these mechanisms remain hypothetical and require rigorous experimental validation. However, they offer a compelling conceptual framework suggesting that ECS signaling may influence EV formation, cargo loading, and release—an emerging interface that may hold significant implications for biomarker discovery and therapeutic innovation in neuropsychiatric disorders.

### 5.2. Extracellular Vesicles as Nanocarriers to Optimize Cannabinoid Pharmacotherapy in Neurological and Psychiatric Conditions

As discussed earlier, cannabinoid-based therapies, particularly those involving CBD and THC, have grown increasing interest for the treatment of neurological and psychiatric disorders due to their anti-inflammatory, neuroprotective, anxiolytic, and antidepressant properties. However, despite promising preclinical results and expanding clinical investigations, these compounds face significant pharmacological limitations that hinder their full therapeutic potential.

One of the primary challenges is their poor oral bioavailability, which in the case of CBD ranges from 6% to 19% in humans, largely due to extensive first-pass hepatic metabolism and low aqueous solubility [159,160]. Consequently, achieving therapeutically effective concentrations in the central nervous system (CNS) often requires high doses, increasing the risk of off-target effects. As for THC, similar oral bioavailability is reported, although for cannabinoids, extensive literature is made on inhalation/pulmonary administration, showing better pharmacokinetic features [161]. For instance, after smoking, THC and CBD bioavailability range 18–50% and 11–45%, respectively [160]. Further details on the pharmacokinetics of cannabinoids, see Lucas and colleagues [162].

Furthermore, cannabinoids have a broad tissue distribution and lack receptor specificity, complicating their application in targeting distinct brain circuits involved in certain illnesses. While CBD is typically well tolerated, excessive quantities of THC can cause dose-dependent adverse effects, such as cognitive impairment, anxiety, paranoia, and transitory psychotic-like symptoms, especially in sensitive groups [163]. These effects not only reduce patient adherence but also raise questions about the long-term safety of cannabinoid-based therapies, particularly in psychiatric illnesses where brain circuits are already functionally damaged.

Another significant restriction is the variable pharmacokinetic profile of cannabis between individuals, which is frequently altered by factors such as age, gender, genetic polymorphisms, food, and concurrent drugs [164]. This heterogeneity hinders dosage standardization and treatment monitoring, resulting in uneven clinical results. Furthermore, traditional administration systems, such as oral capsules, oils, and inhalation, provide little control over the spatial and temporal dynamics of drug release [162]. These issues highlight the need for novel delivery techniques that improve bioavailability, minimize systemic exposure, and ensure focused delivery to the CNS.

In this regard, EVs are emerging as an appealing alternative platform for increasing the pharmacological efficacy of cannabinoid-based medications. Their endogenous origin provides them with superior biocompatibility and low immunogenicity compared to synthetic nanoparticles, reducing the risk of clearance by the immune system and improving biodistribution [165].

Notably, EVs released from neural or immune cells contain surface markers and adhesion molecules that enable their absorption by certain brain cells, including neurons, astrocytes, and microglia, making them suitable vectors for CNS-targeted therapy [166,167]. Moreover, EVs possess the unique ability to traverse the blood–brain barrier (BBB), a major limitation in the development of CNS pharmacotherapies. This capacity has been demonstrated in both animal models and in vitro BBB systems, where EVs successfully delivered therapeutic RNA or small molecules to neuronal targets following systemic administration [168].

EVs offer versatility in drug loading: lipophilic molecules such as cannabinoids can be incorporated into the EV membrane or lumen via passive incubation, sonication, or electroporation, while surface modifications can be employed to improve targeting efficiency [136,169,170]. These characteristics distinguish EVs as a cutting-edge platform for increasing cannabinoids’ bioavailability and CNS penetration, while also lowering systemic exposure and undesirable psychotropic effects.

Preclinical studies exploring the use of extracellular vesicles to enhance the delivery of cannabinoids have demonstrated promising results. Interestingly, a recent study assessed the neuroprotective potential of synthetic CBD and CBD-loaded extracellular vesicles (CBD-EV) isolated from human umbilical cord-derived mesenchymal stem cells in a mouse model of neuropathic pain [171]. The researchers have shown that CBD-EVs were superior to CBD treatment alone in effectively alleviating mechanical and thermal hypersensitivity in a paclitaxel-induced neuropathic pain model.

In a study by Zhu et al. [136], researchers successfully engineered EVs loaded with AM1241, a selective CB2 receptor agonist, to cross the blood–brain barrier and exert neuroprotective effects in an AD mouse model. The authors demonstrated that these EVs could deliver AM1241 specifically to neuronal tissues, where CB2 expression is upregulated under inflammatory conditions, enhancing receptor activation and attenuating neuroinflammation. Notably, treated animals exhibited improvements in cognitive performance, reduced neuroinflammatory markers (such as TNF-α and IL-1β), and restored synaptic integrity, suggesting that CB2-targeted EV delivery may represent a viable strategy to modify AD pathology.

Despite increased interest in the endocannabinoid system (ECS) and EVs in neuropsychiatric research, there have been very few studies that explicitly investigate their interaction. There is a notable lack of evidence on how ECS components, such as endocannabinoids, cannabinoid receptors, or associated enzymes, are transported by EVs in mental illnesses such as anxiety and depression.

Although EVs are now well recognized as significant mediators of intercellular signaling and possible biomarkers in mental disorders, their role in modifying or reflecting ECS activity remains largely unknown. Indeed, to date, no studies have directly evaluated whether EV-mediated delivery of cannabinoids enhances clinical efficacy or alters disease progression in psychiatric models. This represents a critical gap in translational research that warrants focused investigation.

Compared to conventional synthetic nanoparticles, extracellular vesicles offer several advantages that make them particularly suitable for neuropsychiatric applications. First, EVs are inherently biocompatible and exhibit low immunogenicity due to their endogenous origin, reducing the likelihood of immune clearance or adverse systemic responses. Second, EVs display a broad repertoire of surface proteins and lipid components that enable selective targeting and uptake by specific recipient cells, capabilities that are difficult to reproduce in synthetic systems. Third, EVs can be engineered to enhance brain tropism through ligand modification, while still preserving their membrane integrity and cargo stability [172].

In contrast, traditional nanoparticles often require complex chemical functionalization to achieve similar effects, and may still be limited by toxicity, rapid clearance, or inefficient blood–brain barrier penetration. Thus, in the context of cannabinoid-based therapeutics for psychiatric disorders, where precision, safety, and CNS accessibility are paramount, EVs represent a superior and more physiologically compatible delivery platform.

## 6. Future Perspectives

Over the last decades, ECS has emerged as a critical regulator of CNS homeostasis, with essential functions in synaptic plasticity, neuroinflammation, emotional modulation, and stress response. Numerous studies have connected ECS dysregulation to the pathogenesis of a variety of mental and neurological illnesses, including epilepsy, schizophrenia, anxiety, depression, and neurodegenerative diseases.

At the same time, therapeutic strategies targeting the ECS, whether through direct receptor agonists or enzyme inhibitors, have shown promising results in both preclinical and early-phase clinical studies. However, challenges related to pharmacokinetics, receptor selectivity, and safety continue to limit the widespread adoption of cannabinoid-based interventions in clinical practice.

In parallel, EVs have gained increasing recognition as powerful mediators of intercellular communication and potential biomarkers in brain disorders. As discussed earlier, recent discoveries have highlighted the presence of functional endocannabinoids such AEA and 2-AG within EVs derived from microglia and immune cells, suggesting that EVs may participate in the propagation of ECS-related signals under both physiological and pathological conditions. This intersection between EV biology and ECS signaling opens an exciting avenue for research that could transform our understanding of cannabinoid-based pharmacology.

Despite these advances, the integration of EVs and cannabinoid pharmacology remains largely underexplored in the context of psychiatric disorders. While some preclinical studies have investigated EV-mediated delivery of cannabinoids in animal models of neurological disorders, no study to date has directly assessed the therapeutic potential of EV-encapsulated cannabinoids for psychiatric disorders.

Furthermore, the diagnostic potential of ECS-related EV signatures, such as altered endocannabinoid levels or ECS-targeted miRNAs, remains hypothetical. The current literature therefore reveals a significant translational gap that demands rigorous experimental and clinical attention.

Noteworthy, emerging nanotechnological approaches using EVs as delivery systems for cannabinoids offer several potential advantages over conventional formulations. EVs provide a biocompatible, non-immunogenic, and brain-penetrating platform for transporting lipophilic compounds like CBD and THC. Their ability to target specific brain cell populations, protect their cargo from enzymatic degradation, and enable controlled release makes them particularly well-suited for cannabinoid-based therapeutics in complex CNS conditions (Figure 3).

However, major challenges remain before EV-based cannabinoids may be used in clinical settings. These include standardized EV separation and loading techniques, large-scale manufacture, safety profiling, and regulatory approval. Furthermore, more research is needed to improve cannabinoid loading efficiency, characterize pharmacokinetics in vivo, and assess long-term effects in disease-relevant models. A collaborative effort combining neuropharmacology, molecular neuroscience, nanotechnology, and clinical psychiatry will be required to advance this discipline. Special attention should be given to design trials to investigate the therapeutic significance of EV-encapsulated cannabinoids in psychiatric illnesses characterized by ECS dysfunction, such as major depression, anxiety disorders, and schizophrenia.

Building on these observations, we put forward the hypothesis that the ECS–EV interface may represent a novel mechanistic axis relevant to neuropsychiatric vulnerability, resilience, and treatment response. Articulating this hypothesis aims to stimulate systematic investigation into whether EVs can enhance cannabinoid delivery, refine mechanistic understanding, and support the development of more precise, targeted, and clinically meaningful interventions for complex CNS disorders.

## 7. Materials and Methods

This hypothesis-driven review was conducted through a narrative approach, focusing on the interface between cannabinoids and EVs in neuropsychiatric disorders. Relevant literature was searched in PubMed, Scopus, and Web of Science databases up to September 2025. The search strategy combined the following terms and their variations: “endocannabinoid system”, “cannabinoids”, “extracellular vesicles”, “exosomes”, “neuropsychiatric disorders”, “neurodegenerative diseases”, and “psychiatric disorders”. Articles were selected based on their relevance to the proposed hypothesis, with emphasis on preclinical and clinical studies addressing the role of the ECS and extracellular vesicles in neurobiological mechanisms, biomarker identification, and therapeutic strategies. Additional references were retrieved through manual screening of the bibliographies of included articles.

## 8. Conclusions

In conclusion, we propose the hypothesis that the intersection of extracellular vesicle biology and cannabinoid pharmacology constitutes a promising and largely unexplored avenue for advancing the diagnosis and treatment of neuropsychiatric and neurological disorders. By leveraging the unique properties of EVs—including their stability, cell specificity, and capacity to transport bioactive lipids and proteins—future research may overcome key limitations of traditional cannabinoid formulations and enable more precise, effective, and safer therapeutic strategies. Although this field remains in its early stages, articulating this hypothesis aims to catalyze new investigations and deepen our mechanistic understanding of ECS signaling, ultimately fostering innovation in CNS drug delivery and biomarker discovery.

## Figures and Tables

**Figure 1 pharmaceuticals-18-01817-f001:**
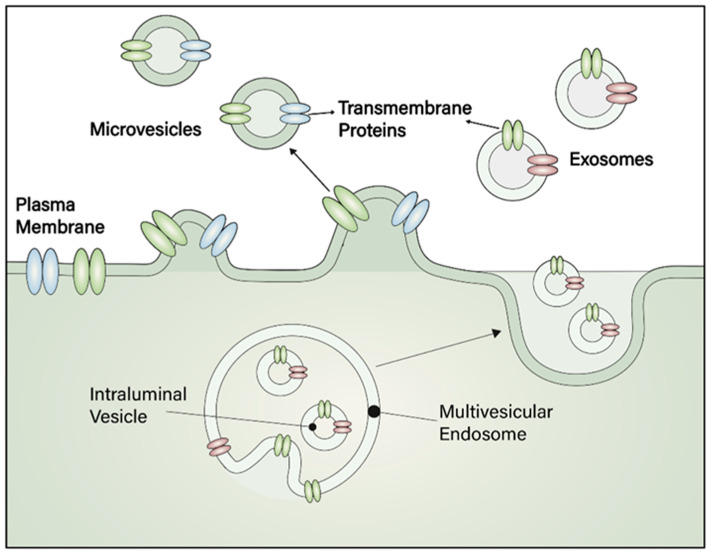
Biogenesis, types, and molecular composition of extracellular vesicles (EVs). On the **top**, a schematic representation illustrates the formation of microvesicles and exosomes. Microvesicles originate from the outward budding of the plasma membrane, whereas exosomes are formed within multivesicular bodies (MVBs) containing intraluminal vesicles and are released into the extracellular space upon fusion with the plasma membrane. On the **bottom**, the molecular composition of EV membranes is depicted, including transmembrane proteins, functional lipids, glycans, and tetraspanins. The intravesicular content comprises nucleic acids (microRNAs, non-coding RNAs, mRNAs, and DNA), biogenesis-related factors (e.g., ALIX, TSG101, syntenin, ubiquitin), signaling proteins (e.g., protein kinases, G-proteins, ARF6, RAB11, ROCK), and enzymes and chaperones (e.g., peroxidases, pyruvate kinase, enolase, GAPDH, HSP70, HSP90). These components highlight the diverse functional roles of EVs in intercellular communication.

**Figure 2 pharmaceuticals-18-01817-f002:**
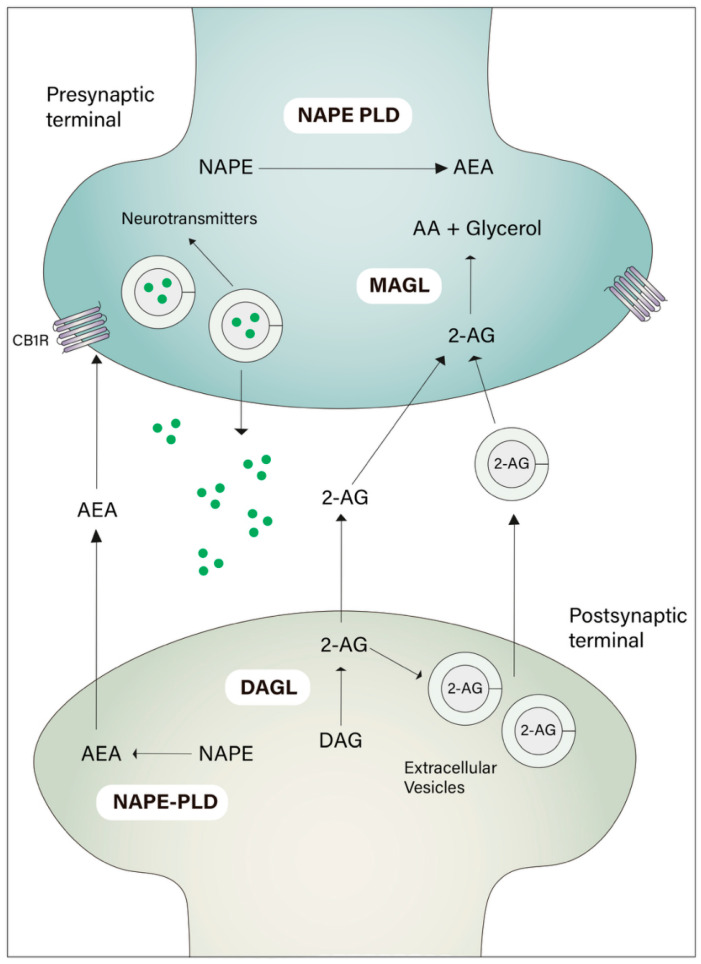
Endocannabinoid system-Extracellular vesicles interface. Endocannabinoid signaling is mediated by stimuli in the postsynaptic neuron. The two major endocannabinoid compounds—Anandamide (AEA) and 2-acyl-glycerol (2-AG)—are synthesized by NAPE-PLD and DAGL, respectively. Classically, both AEA and 2-AG can interact with cannabinoid receptors in presynaptic neurons without the need to be packed in vesicles. However, recent studies have shown the presence of different endocannabinoids in extracellular vesicles released by neuronal cells, highlighting the endocannabinoid-extracellular vesicles interface as an important cellular communication pathway.

**Figure 3 pharmaceuticals-18-01817-f003:**
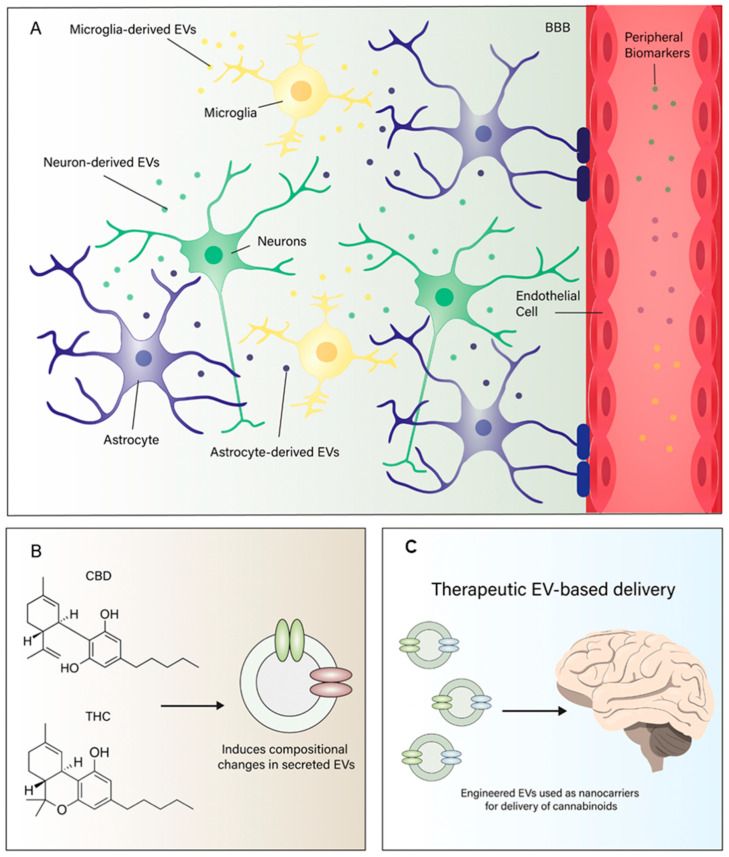
(**A**) Central nervous system (CNS) cells—including neurons, astrocytes, microglia, and oligodendrocytes—release extracellular vesicles (EVs) that carry molecular cargo such as endocannabinoids (e.g., anandamide, AEA), regulatory proteins, and microRNAs (miRNAs). These EVs can cross the synaptic cleft and influence neuronal and glial targets by modulating synaptic activity, neuroinflammation, and glia–neuron communication through CB1, CB2, and other receptor systems. (**B**) Exposure to phytocannabinoids—such as Δ9-tetrahydrocannabinol (THC) and cannabidiol (CBD)—as well as synthetic cannabinoids modulates EV biogenesis and cargo composition, influencing intercellular communication and potentially reprogramming neural and immune signaling. (**C**) Therapeutic EV-based delivery: engineered EVs can be used as nanocarriers for targeted CNS delivery of cannabinoids, increasing brain bioavailability, enhancing cell-type selectivity, and minimizing systemic side effects. This bidirectional interaction between the endocannabinoid system (ECS) and EVs represents a novel translational platform for the development of precision neuropsychiatric treatments.

**Table 1 pharmaceuticals-18-01817-t001:** Studies supporting the use of EVs as biomarkers and/and therapeutic tools in the context of neuropsychiatric disorders.

Neuropsychiatric Disorder	EV Isolation Method/ Quality Method/Analysis	Biomarker/ Therapeutic	Key Results	References
**Alzheimer’s disease**	dUC of conditioned medium	Biomarker/ Pathogenesis	Microglial Aβ-EVs significantly affect synaptic plasticity both in cultured neurons and in brain slices. When injected into mouse brains, these EVs propagate synaptic dysfunction in the entorhinal-hippocampal circuit.	[125]
**Parkinson’s disease**	dUC/immunoaffinity for L1CAM in human serum	Biomarker	Neuron-derived extracellular vesicles contained elevated α-synuclein levels in both patients with Parkinson’s disease and individuals who later developed the disorder, effectively distinguishing future converters from controls and supporting their value as predictive and diagnostic biomarkers.	[126]
**Parkinson’s disease**	Size-exclusion chromatography from human serum	Biomarker	Profiling of EV-derived microRNAs identified distinct expression signatures that accurately discriminated healthy controls, patients with idiopathic REM sleep behavior disorder, and PD, with specific miRNA panels predicting conversion from iRBD to PD, highlighting their potential as early, minimally invasive biomarkers.	[127]
**Parkinson’s disease**	dUC from both conditioned medium and human serum	Biomarker	Plasma-derived exosomes from PD patients contained elevated monomeric and oligomeric α-synuclein, preferentially entered microglia (rather than neurons or astrocytes) in vivo and in vitro, activated microglial inflammatory responses (NO, TNF-α, IL-6). In mice, these exosomes induced α-synuclein pathology and dopaminergic neuron dysfunction in striatum.	[128]
**Schizophrenia/** **Bipolar disorder**	dUC of frozen human prefrontal cortex	Biomarker	Exosome-containing pellets from patients with schizophrenia showed significantly increased expression of miR-497, and from bipolar disorder patients significantly increased miR-29c, compared to controls, suggesting disease-specific exosomal miRNA signatures in prefrontal cortex tissue.	[129]
**Schizophrenia**	Size-exclusion chromatography	Biomarker	Identified five differentially expressed surface proteins (FN1, ITGB3, PECAM1, ITGA6, CD5) that distinguished first-episode schizophrenia patients from healthy controls (AUC up to ~0.805), and six proteins (ITGB3, CD33, CD40, CD36, TENM2, EGFR) that distinguished antipsychotic non-responders from responders (AUC up to ~0.87).	[130]
**Schizophrenia**	Total Exosome Isolation Reagent (Invitrogen/Thermofisher) from serum	Biomarker	Exosomal GFAP concentration was significantly higher and α-II Spectrin expression significantly lower in plasma from schizophrenia patients compared with matched healthy controls, indicating a peripheral signature of astrocytic pathology in schizophrenia.	[131]
**Autism Spectrum Disorder**	Size-exclusion chromatography from serum	Biomarker	Proteomic profiling of EVs revealed that five proteins (WWP2, HSP27, CLEC1B, CD40 and FRα) were significantly down-regulated in plasma EVs from individuals with autism spectrum disorder compared with healthy controls (fold-change ≥ 2, adjusted *p*-value ≤ 0.05), and a machine-learning model based on these proteins achieved an AUC ≈ 0.923 for classification of ASD vs. controls.	[132]
**Autism Spectrum Disorder**	exoEasy Maxi Kit (Qiagen) (Membrane-affinity spin column)	Biomarker	Children with autism spectrum disorder had significantly increased total EV--EV-associated protein and mitochondrial DNA (mtDNA7S) in serum compared to normotypic controls; furthermore, when applied to human microglia, these serum EVs stimulated IL-1β secretion in a dose- and time-dependent manner, supporting a potential pro-inflammatory role for EVs in ASD.	[133]
**Major Depressive Disorder**	dUC of human serum	Biomarker/ Pathogenesis	Exosomes from MDD patients carried elevated levels of miR-139-5p, injected into healthy mice, induced depressive-like behaviors, while exosomes from healthy volunteers alleviated stress-induced depressive behavior; exosomal miR-139-5p was shown to inhibit hippocampal neurogenesis and may mediate depression pathophysiology.	[134]
**Stress-related disorders**	dUC followed by size-exclusion chromatography	Biomarker	Proteomic and transcriptomic profiling of EVs revealed distinct signatures associated with neurodevelopmental and psychiatric disorders. Specific EV cargo alterations correlated with synaptic and inflammatory pathways, suggesting that circulating EVs may serve as biomarkers reflecting central nervous system dysfunction.	[135]
**Alzheimer’s disease**	dUC followed by size-exclusion chromatography	Therapeutic	EVs loaded with the CB2 agonist AM1241 reversed neurodegenerative pathology and enhanced neurogenesis in mouse models, indicating that engineered EVs can serve as effective therapeutic carriers in neurodegenerative disease.	[136]

Abbreviations: dUC—Differential Ultracentrifugation; ASD—Autism Spectrum Disorder; EV—Extracellular Vesicles; PD—Parkinson’s disease.

## Data Availability

No new data were created or analyzed in this study. Data sharing is not applicable to this article.

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
