# Peer review of "Extracellular Vesicles as a Potential Biomarker and Therapeutic Opportunity for Neuropsychiatric Disorders: A Hypothesis-Driven Review"

_pharmaceuticals, 2025, doi:10.3390/ph18121817_

Round 1
Reviewer 1 Report
Comments and Suggestions for Authors
The content of the review does fulful the enthusiasm provoked by the title.
Section 1 is reasonable and set the stage effectively.
Sections 2-4 are not novel and thus do not advance the field.
Section 5 is interesting but very little information is provided. This is likely due to gaps in the literature, but this is the area that the title suggested would be covered.
What exactly is the hypothesis that "drives" this review?
Biomarker information is lacking
Author Response
The content of the review does fulful the enthusiasm provoked by the title.
We thank you the reviewer for the kind words regarding the content of the review.
Section 1 is reasonable and set the stage effectively.
Sections 2-4 are not novel and thus do not advance the field.
Thank you very much for your observation. Indeed, these two sections are not new, and the literature already includes excellent reviews and studies on these topics. However, because our intention is to stimulate new research in both areas—cannabinoids and EVs in the context of neuropsychiatric disorders—we decided to keep this content. We believe it may be useful for EV specialists who are not yet familiar with the cannabinoid field in neuropsychiatric disorders, providing them with a concise overview of the current state of the art.
Section 5 is interesting but very little information is provided. This is likely due to gaps in the literature, but this is the area that the title suggested would be covered.
Biomarker information is lacking
Thank you very much for your observations and suggestions.
While we maintained Section 5 (now renumbered as Section 6 in the revised manuscript), we have incorporated two new sections to address the reviewers’ suggestions and to strengthen the mechanistic and translational aspects of the manuscript.
First, we added a section dedicated to biomarkers in neuropsychiatric disorders (pp. 9–10): 3.0 – Biomarkers in the Context of Neuropsychiatric Disorders. This section consolidates current evidence and highlights the relevance of molecular and cellular biomarkers, including those derived from EVs in the diagnosis, prognosis, and stratification of neuropsychiatric conditions.
Finally, and importantly, we incorporated a mechanistic discussion section: 5.1 – Possible Mechanisms Involved in the Interface Between EVs and Cannabinoids. This new section directly addresses the reviewer’s request for deeper mechanistic insights, summarizing potential pathways through which cannabinoids may modulate EV biogenesis, release, and cargo loading within neural and immune systems.
What exactly is the hypothesis that "drives" this review?
Thank you for your important observation. To make the hypothesis of the reviewer clearer, we have now added a paragraph that states our hypothesis
Page 3 lines 106 to 116
“In this context, in the present review, we introduce our hypothesis that the interface between cannabinoids and extracellular vesicles (EVs) represents a critical and underexplored axis in the pathophysiology of neuropsychiatric disorders. Specifically, we propose that cannabinoid–EV interactions may serve as a source of novel biomarkers and therapeutic targets, offering a promising avenue for improving the diagnosis, monitoring, and personalization for the treatment of these conditions. By elucidating how cannabinoids modulate EV biogenesis, cargo, and signaling, this framework may uncover key molecular and cellular mechanisms underlying neuropsychiatric dysfunctions, thereby helping to overcome the persistent limitations of current pharmacological interventions and ultimately improve clinical endpoints by promoting treatment adherence via personalized medicine. “
We sincerely thank the reviewers for their time, care, and thoughtful consideration in providing such constructive comments and suggestions. All changes made in the main document are highlighted in red.
Reviewer 2 Report
Comments and Suggestions for Authors
This review work contains innovative hypotheses and interesting information involving cannabinoids and extracellular vesicles in neuropsychiatric disorders. However, I have some comments and suggestions:
- I suggest to add more detail discussion of mechanistics explanation of how cannabinoids might influence the release in neuronal systems, cargo loading, and EV biogenesis.
- Highlight the global problem (health and economic) of neuropsychiatric disorders by adding a paragraph in the introduction about the WHO's latest statistics on neuropsychiatric disorders.
- The schematic figures are clear and well-constructed.
- The references section is well written and covered, even with updated citations (2025).
- In general, the review manuscript in all its parts is well designed, well written and I suggest minor language polishing for the whole manuscript is needed.
Minor language polishing required
Author Response
This review work contains innovative hypotheses and interesting information involving cannabinoids and extracellular vesicles in neuropsychiatric disorders. However, I have some comments and suggestions:
- I suggest adding more detail discussion of mechanistics explanation of how cannabinoids might influence the release in neuronal systems, cargo loading, and EV biogenesis.
Thank you very much for your suggestion. To address your comment, we added a new section dedicated to biomarkers in neuropsychiatric disorders (pp. 9–10): 3.0 – Biomarkers in the Context of Neuropsychiatric Disorders.
Finally, and importantly, we incorporated a mechanistic discussion section: 5.1 – Possible Mechanisms Involved in the Interface Between EVs and Cannabinoids. This new section directly addresses the reviewer’s request for deeper mechanistic insights, summarizing potential pathways through which cannabinoids may modulate EV biogenesis and release.
Highlight the global problem (health and economic) of neuropsychiatric disorders by adding a paragraph in the introduction about the WHO's latest statistics on neuropsychiatric disorders.
Thank you very much for your suggestion. We have now included the WHO data in the first paragraph of the review:
“According to the World Health Organization (WHO), 1 of every people (~1 billion) in world are currently living with a neuropsychiatric disorders, including both mental and neurological conditions”
- The schematic figures are clear and well-constructed.
- The references section is well written and covered, even with updated citations (2025).
Thank you very much for your positive comments. We appreciated you input.
- In general, the review manuscript in all its parts is well designed, well written and I suggest minor language polishing for the whole manuscript is needed.
Thank you very much. We will use language editing services of the MPI to improve the language of the revised version of the review.
We sincerely thank the reviewers for their time, care, and thoughtful consideration in providing such constructive comments and suggestions. All changes made in the main document are highlighted in red.
Reviewer 3 Report
Comments and Suggestions for Authors
The review discusses how ECS dysregulation is implicated in many CNS disorders and explores the potential of extracellular vesicles (EVs) as biomarkers and cannabinoid carriers for diagnosis and therapeutics. It highlights the ability of EVs to transport bioactive molecules, including endocannabinoids, across the blood–brain barrier, offering advantages such as improved drug delivery, enhanced brain targeting, and reduced systemic side effects. The authors propose EVs as nanocarriers to optimize cannabinoid pharmacotherapy by addressing challenges such as poor bioavailability and receptor specificity. Overall, the manuscript is well written and covers a timely and important research topic regarding the roles of EVs in cannabinoid-related disorders and therapeutics. However, the review provides limited evidence to support the hypothesis that EVs may serve as reliable biomarkers or pharmacological tools with diagnostic and therapeutic potential.
Major Concerns:
- Section 3 is too long and somewhat off-topic. It should be substantially shortened, as the involvement of the ECS in CNS disorders has already been extensively reviewed in the literature. The authors should instead focus more on the roles of EVs in diagnostics and therapeutics.
- Findings on EVs as biomarkers are scattered, making it difficult to identify key studies and conclusions. The authors should better integrate and discuss ECS-related EV signatures - such as altered endocannabinoid levels or ECS-targeted miRNAs - as potential biomarkers for early detection and risk stratification in neuropsychiatric disorders.
- Use summary tables to present key studies that support the roles of EVs as biomarkers versus therapeutic tools, thereby improving clarity and accessibility.
- Further research is needed on the molecular mechanisms underlying the ECS–EV interface, particularly in psychiatric disorders, to better understand how EVs influence ECS signaling.
Author Response
The review discusses how ECS dysregulation is implicated in many CNS disorders and explores the potential of extracellular vesicles (EVs) as biomarkers and cannabinoid carriers for diagnosis and therapeutics. It highlights the ability of EVs to transport bioactive molecules, including endocannabinoids, across the blood–brain barrier, offering advantages such as improved drug delivery, enhanced brain targeting, and reduced systemic side effects. The authors propose EVs as nanocarriers to optimize cannabinoid pharmacotherapy by addressing challenges such as poor bioavailability and receptor specificity. Overall, the manuscript is well written and covers a timely and important research topic regarding the roles of EVs in cannabinoid-related disorders and therapeutics. However, the review provides limited evidence to support the hypothesis that EVs may serve as reliable biomarkers or pharmacological tools with diagnostic and therapeutic potential.
Major Concerns:
- Section 3 is too long and somewhat off-topic. It should be substantially shortened, as the involvement of the ECS in CNS disorders has already been extensively reviewed in the literature. The authors should instead focus more on the roles of EVs in diagnostics and therapeutics.
Thank you very much for your observation. Indeed, literature already includes excellent reviews and studies on these topics. However, because our intention is to stimulate new research in both areas—cannabinoids and EVs in the context of neuropsychiatric disorders—we decided to keep this content (we have reduced it as suggested). We believe it may be useful for EV specialists who are not yet familiar with the cannabinoid field in neuropsychiatric disorders, providing them with a concise overview of the current state of the art.
Additionally, we have incorporated two new sections in the review that we believe is aligned with your comments:
3.0 – Biomarkers in the Context of Neuropsychiatric Disorders.
And,
5.1 – Possible Mechanisms Involved in the Interface Between EVs and Cannabinoids.
- Findings on EVs as biomarkers are scattered, making it difficult to identify key studies and conclusions. The authors should better integrate and discuss ECS-related EV signatures - such as altered endocannabinoid levels or ECS-targeted miRNAs - as potential biomarkers for early detection and risk stratification in neuropsychiatric disorders.
Thank you very much. We agree with the reviewer, and we believe we have addressed your important observation in the new session 3.0 – Biomarkers in the Context of Neuropsychiatric Disorders.
- Use summary tables to present key studies that support the roles of EVs as biomarkers versus therapeutic tools, thereby improving clarity and accessibility.
Thank you. A table (1) was added as suggested.
- Further research is needed on the molecular mechanisms underlying the ECS–EV interface, particularly in psychiatric disorders, to better understand how EVs influence ECS signaling.
Thank you for your suggestion. We have now added a topic 5.1 – Possible Mechanisms Involved in the Interface Between EVs and Cannabinoids, covering this subject.
We sincerely thank the reviewers for their time, care, and thoughtful consideration in providing such constructive comments and suggestions. All changes made in the main document are highlighted in red.
Round 2
Reviewer 1 Report
Comments and Suggestions for Authors
The authors have adequately addressed the initial concerns.
Author Response
The authors have adequately addressed the initial concerns.
We sincerely thank the reviewers for the time and careful attention dedicated to evaluating our manuscript.
Reviewer 2 Report
Comments and Suggestions for Authors
I would like to thank the authors for their hard work and the well-constructed corrections made to the manuscript. I have no further comments or suggestions.
Author Response
I would like to thank the authors for their hard work and the well-constructed corrections made to the manuscript. I have no further comments or suggestions.
We sincerely thank the reviewer for the time and careful attention dedicated to evaluating our manuscript.
Reviewer 3 Report
Comments and Suggestions for Authors
Although the authors have addressed some of my previous comments, after carefully reviewing the revised manuscript I still do not find solid evidence supporting the central hypothesis emphasized in the title and abstract. As currently written, the title and content are not fully aligned and may even be somewhat misleading. I recommend that the authors revise the title to something more accurate, for example, ‘Extracellular Vesicles as a Potential Biomarker and Therapeutic Opportunity for Neuropsychiatric Disorders or Cannabis Use Disorders" or a similar phrasing. In addition, the statements regarding the uncertain "cannabinoid-EV interface" in the abstract should be softened to avoid overstating such an interface that is not supported by the presented evidence.
Author Response
Although the authors have addressed some of my previous comments, after carefully reviewing the revised manuscript I still do not find solid evidence supporting the central hypothesis emphasized in the title and abstract. As currently written, the title and content are not fully aligned and may even be somewhat misleading. I recommend that the authors revise the title to something more accurate, for example, ‘Extracellular Vesicles as a Potential Biomarker and Therapeutic Opportunity for Neuropsychiatric Disorders or Cannabis Use Disorders" or a similar phrasing. In addition, the statements regarding the uncertain "cannabinoid-EV interface" in the abstract should be softened to avoid overstating such an interface that is not supported by the presented evidence.
Thank you very much for your valuable suggestions and for the time dedicated to evaluating our manuscript.
We have changed the title of the manuscript according to the reviewer’s suggestion. It now reads:
"Extracellular Vesicles as a Potential Biomarker and Therapeutic Opportunity for Neuropsychiatric Disorders: A Hypothesis-Driven Review."
Additionally, we have rewritten the sentence highlighted by the reviewer as follows:
"We propose a hypothesis-driven framework in which the possible interplay between cannabinoids and EVs may stimulate new research and support the development of biomarker-guided, personalized therapeutic strategies for neuropsychiatric disorders."